# Relationships of a Detailed Mineral Profile of Meat with Animal Performance and Beef Quality

**DOI:** 10.3390/ani9121073

**Published:** 2019-12-03

**Authors:** Nageshvar Patel, Matteo Bergamaschi, Luciano Magro, Andrea Petrini, Giovanni Bittante

**Affiliations:** 1Department of Agronomy, Food, Natural Resources, Animals and Environment (DAFNAE), University of Padova, viale dell’Università 16, 35020 Legnaro (PD), Italy; luciano.magro@unipd.it (L.M.); bittante@unipd.it (G.B.); 2Department of Animal Science, North Carolina State University, Raleigh, NC 27695, USA; mbergam@ncsu.edu; 3Consorzio Tutela del Vitellone Bianco dell’Appennino Centrale, via delle Fascine 4, San Martino in Campo, 06132 Perugia, Italy; petrini@vitellonebianco.it

**Keywords:** macro-minerals, micro-minerals, environmental-minerals, beef quality, beef production, multivariate analysis

## Abstract

**Simply Summary:**

Minerals play direct or indirect role in different biological process of animals. These biological processes finally affect the meat quality. Therefore, analysis of minerals in cattle’s diet is important for assessing potential effects on beef quality. However, minerals profile and concentration in beef are affected by several factors such as animals rearing practices, age, environment, breed etc. Hence, we analyzed 20 minerals in 192 beef samples and studied the different sources of variation which affect the minerals profile in beef. In order to understand the complex and intriguing relations of beef qualities and minerals, we utilized correlation and factor analysis with 16 traits related to animal performance and beef quality. Our analysis shows many significant associations of minerals in beef with animal performance and beef qualities. Five groups of minerals (latent factors) were associated with almost all quality traits of beef. The knowledge about the mineral contents in beef is important to understand the complex interrelationships of animal rearing, farm management, environmental conditions with regard to animal performance and beef quality.

**Abstract:**

The mineral profile of beef is a subject of human health interest, but also animal performance and meat quality. This study analyzes the relationships of 20 minerals in beef inductively coupled plasma-optical emission spectrometry (ICP-OES) with three animal performance and 13 beef quality traits analyzed on 182 samples of *Longissimus thoracis*. Animals’ breed and sex showed limited effects. The major sources of variation (farm/date of slaughter, individual animal within group and side/sample within animal) differed greatly from trait to trait. Mineral contents were correlated to animal performance and beef quality being significant 52 out of the 320 correlations at the farm/date level, and 101 out of the 320 at the individual animal level. Five latent factors explained 69% of mineral co-variation. The most important, “*Mineral quantity*” factor correlated with age at slaughter and with the beef color traits. Two latent factors (“*Na + Fe + Cu*” and “*Fe + Mn*”) correlated with performance and beef color traits. Two other (“*K-B-Pb*” and “*Zn*”) correlated with beef chemical composition and the latter also with carcass weight and daily gain, and beef color traits. Beef cooking losses correlated with “*K-B-Pb*”. Latent factor analysis appears be a useful means of disentangling the very complex relationships that the minerals in beef have with animal performance and beef quality traits.

## 1. Introduction

Minerals are valuable nutrients and essential for both human and animal health [1]. Minerals are known to have a large influence on human health, effecting a wide variety of body functions, for example, enzyme function, osmotic pressure control, muscle contraction, etc. [2,3,4]. Minerals also plays crucial role in meat quality because they affect the several biological process in animals and some characteristics of beef such as color and texture. Therefore, minerals are relevant, so their characterization in beef and exploring their relation with animal performance and beef quality becomes important. However, as several studies have shown that type and concentrations of minerals in beef are affected by several factors, such as the animal’s breed, sex, age, diet and water intake, farm management system, environmental conditions [5,6,7], muscle type and cooking procedure [8]. The mineral content of beef has been studied in particular in relation to its value as a nutrient for humans [9,10] and, although less frequently, in relation to the adequacy of the animals’ dietary mineral supply [1]. Although several studies have quantified the minerals contained in beef, most have looked at only a few of them—macro-minerals and some essential micro-minerals—and have often sampled only one experimental farm with one feeding regime. 

Furthermore, with a few exceptions, such as the relationships between iron content and beef color [11], and between calcium content and beef tenderness [12,13], almost no studies have systematically analyzed the relationships between mineral content and beef quality traits (meat chemical composition, color, texture, cooking losses, etc.).

As we analyzed fairly large numbers of minerals, productive and qualitative traits, we did not set out to examine in detail and discuss each single mineral and its relationships with all the animal and beef traits, but rather aimed to obtain a general picture of the main associations between the detailed mineral profile and animal performance and beef quality. The specific objectives of this investigation, therefore, were: (a) to analyze firstly the sources of variation in animal performance and beef quality traits (breed, sex, farm, animal within farm, sample within animal) in animals reared on 15 different farms in accordance with a recognized beef production system, which we treated as a case study; (b) to analyze the relationships between 20 mineral concentrations in beef (six macro-minerals, five essential micro-minerals, and nine environmental micro-minerals) and animal performance and beef quality traits at the farm level and also at the level of individual animal within farm; and (c) to analyze the relationships of the latent explanatory factors condensing the major part of the co-variation in the minerals with animal performance and beef quality as a means of identifying possible pathways characterizing the complex picture emerging from the correlations. 

## 2. Materials and Methods

### 2.1. Farms and Animals 

The beef production system that serves as our case study has Protected Geographical Indication (PGI) certification under European Union regulation 134/1998 with the designation “Vitellone Bianco dell’Appennino Centrale” (Central Apennine White Young Bull). Fifteen farms in the historical areas of origin of the Chianina (mainly Tuscany/Umbria) and the Romagnola (Emilia-Romagna) breeds, were selected by the “Consorzio Produttori Carne Bovina Pregiata delle Razze Italiane” (CCBI, Consortium of Producers of High-Quality Beef from Italian Breeds), which is responsible for controlling and monitoring the PGI certification. Ninety-one young bulls and heifers of the two breeds were randomly sampled from the 15 selected farms. According to PGI regulations, calves remain with the suckler cows till weaned, often at pasture from spring to fall, and are fattened with traditional feeding practices based on forages and concentrates (compound feed and/or cereal mix). Silages are prohibited during the last two months of fattening. Slaughter date is decided by the farmer for each animal individually according to local market requirements, i.e., at carcass fatness scores of about 2.0 for young bulls and 2.5 for young heifers [14] on the European SEUROP carcass classification system [15].

Ninety one animals (83 young bulls, eight heifers; 39 Chianina, 52 Romagnola) were registered in the Herd Books of their respective breeds, which are managed by the Associazione Nazionale Allevatori di Bovini Italiani da Carne (ANABIC; National Association of Italian Beef Cattle Farmers, Perugia, Italy). The low proportion of heifers reflect the market availability as the large majority of heifers are used for replacing culled cows. They were sired by 11 Chianina and 35 Romagnola bulls (mainly through artificial insemination), representative of the current genetics of these beef breeds. The animals’ ages, carcass weights, and daily carcass weight gains are presented in Table 1.

### 2.2. Beef Samples

Animals were slaughtered in accordance with European Union regulations [16], and the carcass weight recorded. The day after slaughter, sample beef joints were obtained from the *longissimus thoracis* muscle at the level of the division of the carcass sides into two quarters according the pistol cut (5th rib). Both sides of each carcass (182 beef samples from 91 animals) were sampled separately in order to assess the effects on the quality of beef sampled from the same anatomical position in two sides of the same animal. Samples were cooled, vacuum packed, and labeled, then taken to the DAFNAE Beef Laboratory, University of Padova, Italy, for analysis.

### 2.3. Analysis

#### 2.3.1. Beef Quality Analysis

After 7 days of aging at 4 °C, the pH of each of the *Longissimus thoracis* samples was measured at three different points with a Delta Ohm HI-8314 pH meter. The average value of the three replicates was used for the subsequent analyses. The sample beef joints were then cut perpendicularly to the muscle fibers, and one hour later the color of the muscle surface was measured at five different points using a Minolta CM-508c (illuminate: D65, observer: 100) according to the procedure described by [17,18]. Mean color was expressed in *L*, a*, b*, C** and *h** values [19]. 

To measure cooking losses, a 2cm-thick beef steak was placed in a polyethylene bag and cooked in a water bath at 70 °C for 40 min. Cooking loss was calculated as the percentage difference between the weight of the beef before and after cooking [20]. The texture of the cooked beef sample was measured by shear force using a TA-HDi Texture Analyser (Stable Micro Systems, Godalming, UK) with a Warner Bratzler shear attachment (10N load cell, crosshead speed of 2 mm/s) and analyzed with the Texture Expert software (Stable Micro Systems, Godalming, UK) [21]. The average of three replicates was used for the beef texture analysis. 

Chemical analysis of the beef was carried out according to [22]. Ash content was measured after drying the beef at 525 °C, according to the AOAC method, and lipid percentage was determined by the extraction method using petroleum ether. Cholesterol content was measured by extracting it by saponification and following the method described by {Formatting Citation}.

#### 2.3.2. Mineral Analysis

The procedure used for analyzing the mineral content of the beef samples was reported in detail in a previous paper [23]. Briefly, after 7 days of aging at 4 °C, fat from the outer side of each beef sample was inspected and trimmed. A sub sample of the beef was ground then freeze-dried. 

The beef samples were analyzed for their mineral contents, and after identification the various minerals were quantified with a Spectro Arcos EOP ICP-OES (Spectro A.I. GmbH, Kleve, Germany). All instrument operating parameters were optimized for nitric acid 30% solution. Between 0.300 and 0.350 g of freeze-dried tissue from each sample was placed in a TFM vessel with 2 mL of 30% hydrogen peroxide and 7 mL of concentrated (65%) nitric acid, both Suprapur^®^ quality (Merck Chemicals GmbH, Darmstadt, Germany). These prepared samples were subjected to microwave (Ethos 1600, Milestone S.r.l., Sorisole, BG, Italy) digestion as follows: Step 1, 25–200 °C in 15 min at 1200 W with P max 100 bar; Step 2, 200 °C for 15 min at 1200 W with P max 100 bar; Step 3, 200–110 °C in 15 min. After cooling down to room temperature, the dissolved sample was diluted with ultrapure water (resistivity 18.2 M Ω cm at 25 °C) to a final volume of 25 mL. 

Calibration standards were prepared using multi element and single element standard solutions (Inorganic Ventures Inc., Christiansburg, VA, USA) in 18% Suprapur^®^ nitric acid to obtain similar matrices to the samples. Concentrations of 0, 0.005, 0.02, 0.05, 0.2, 0.5, 2 and 5 mg/L of the analytes were prepared. The calibration solutions for calcium, potassium, magnesium, sodium, phosphorous and sulfur were at the same concentrations as the other analytes plus further concentrations of 20, 50 and 200 mg/L. 

Method precision and trueness were assessed with a blank solution, a low-level control solution (recovery limits ± 30%), a medium-level control solution (recovery limits ± 10%), and the international standard reference material NIST SRM 1577c (National Institute of Standards & Technology (NIST), Gaithersburg, MD, USA), prepared as described above. 

A total of 20 minerals (Table 2), comprising six essential macro-minerals (Na, Mg, P, S, K, and Ca), 5 essential micro-minerals (Cr, Mn, Fe, Mn, and Zn) and nine environmental micro-minerals (Li, B, Al, Ni, Sr, Sn, Ba, Ti, and Pb) were present in the beef samples in quantities above our limit of quantification (LOQ). A further 10 minerals (As, Be, Cd, Co, Hg, Mo, Sb, Se, Ti, and V) were also identified, but these were not present in at or above the LOQ in all the beef samples. The halogens (Cl, I, Br) were not identified as these minerals need special sample preparation and are detected at very low wavelengths, which require the instrument parameters to be reset with subsequent loss of precision. It is of note that the coefficients of variation in beef were very different for different minerals [23]. In the case of the essential macro-minerals it was modest, ranging from 3% for K to 11% for Na, with the exception of Ca (28%). The coefficients of variation of the essential micro-minerals were in the range 13–17%, with the exception of Cr (51%), whereas those of the environmental micro-minerals were much larger, 35 to 64%, with the exception of Pb (22%).

### 2.4. Statistical Analysis

#### 2.4.1. Mixed Model Analysis of Variance

All data were normally distributed or very close to normality (Q-Q probability plot and Shapiro-Wilks test). A comprehensive statistical analysis of variance of the data on animal performance and beef quality traits was then carried out using the PROC MIXED procedure in SAS version 9.4 (SAS Institute Inc., Cary, NC, USA) to quantify the fixed (breed, sex) and random (farm/date, animal within farm/date, sample/side within animal) sources of variation of the traits. The following model was used for the analysis of variance:y_ijkl_ = μ + breed_i_ + farm/date(breed)_i:j_ + sex_k_ + animal (sex)_k:l_ + e_ijkl_
where y_ijkl_ is the trait studied (animal performance and beef quality traits); μ is the overall mean; breed_i_ is fixed effect of breed (i = Chianina, Romagnola); farm/date (breed)_i:j_ is the random effect of the jth farm/slaughter date within breed (j = 1–15), which was used to test the significance of the breed effect; sex_k_ is the fixed effect of sex (k = male, female); animal (sex)_k:l_ is the random effect of the lth animal within sex (l = 1–91) which was used to test the significance of sex effect; e_ijkl_ is the residual random error term referring to the differences between the beef samples taken from the two sides of each carcass, ~N(0,σ^2^), where σ^2^ is the side/residual variance. As age at slaughter, carcass weight and carcass daily gain had only one value per animal, they were analyzed with a reduced model, similar to the previous one but excluding the animal (sex) effect, so that the residual term also summarizes the animal effect.

Tests for outliers were based on the residuals yielded by the above described model in a preliminary run for all the animal performance and beef quality traits. Data whose residuals were within the range of ± 3 residual standard deviations were kept, while the rest were treated as outliers and therefore omitted from final analysis.

#### 2.4.2. Correlations and Multivariate Statistical Analysis

Farm/date and animal within farm/date correlations were analyzed to identify the relationships of the mineral contents to the beef quality traits and the animals’ phenotypic traits, which yielded 640 correlation coefficients: 20 minerals × 16 animal traits (three performance traits, five beef composition traits, three beef physical quality traits, and five color traits) × 2 types of correlations (correlations among herd/date solutions and among animals within herd/date).

A multivariate analysis to capture the major part of the co-variation among the minerals was carried out and reported in a previous study (see Patel et al., [23] for details). Factor analysis was carried out in 3 steps using Varimax rotation in the R studio environment version 3.4.1 using the psych package. The eigenvalues of the factors and the communality values for the measured variables after rotation were also obtained. Five unmeasured latent explanatory independent factors were identified, explaining 69% of the total co-variation among the 20 minerals. Firstly, KMO (Kaiser-Meyer-Olkin) and Bartlett’s test were carried out and confirmed that the mineral data were suitable for factor analysis.

The five unmeasured latent explanatory independent factors were:(1)Factor “*Quantity*”: Eigen value 4.9, representing 45.2% of the co-variation explained by all factors, related to the beef content of P (loading 0.96), S (0.74), Mg (0.68), Cr (0.68), Al (0.64), Ti (0.71), Pb (0.78), Ba (0.53), and Sn (−0.74);(2)Factor “*Na + Fe + Cu*”: Eigen value 2.2, representing 17.9% of the co-variation explained by all factors, related to the beef content of Na (loading 0.66), Fe (0.77), and Cu (0.60);(3)Factor “*K-B-Pb*”: Eigen value 1.7, representing 15.6% of the co-variation explained by all factors, related to the beef content of K (loading 0.76), B (−0.53), and Pb (−0.54);(4)Factor “*Fe + Mn*”: Eigen value 1.2, representing 10.8% of the co-variation explained by all factors, related to the beef content of Fe (loading 0.62), and Mn (0.48);(5)Factor “*Zn*”: Eigen value 1.1, representing 10.4% of the co-variation explained by all factors, related to the beef content of Zn (loading 0.94).

The scores of each latent factor associated with the minerals in each beef sample were used to calculate the coefficients of the correlations of farm/date and animal within farm/date with animal performance and beef quality traits.

## 3. Results 

### 3.1. Animal Performance and Beef Quality and Their Sources of Variation

Descriptive statistics of animal performance and beef quality traits, and the significance levels of the fixed effects included in the model are presented in Table 1. The results show that the only differences due to breed that reached significance were in cooking losses, the shear force of cooked beef, and the lightness of raw beef. The Chianina beef was lighter than the Romagnola beef (L* 33.5 vs. 36.8), but showed greater cooking losses (35.9 vs. 32.5%) and shear force (31.9 vs. 28.6 N/cm^2^).

There were no differences in the beef chemical compositions of the two beef breeds, whereas the beef from the females had, as expected, greater dry matter and lipid contents than the beef from the males (27.7 vs. 25.9%, and 3.93 vs. 2.20%, respectively). 

Analysis of the sources of variation treated as random factors, Figure 1 depicts the relative importance of group (farm/date of slaughter) and individual animal (carcass) within group to the total variance in beef performance traits, and of beef sample/side within animal (carcass) to the total variance in beef quality traits. It shows that the variability in all performance traits (age at slaughter, carcass weight and daily gain) is more affected by differences among groups (about two thirds of total variance) than by differences among animals within group (one third).

Beef quality traits, such as dry matter, lipids, color traits (excluding *L**), and cooking losses were highly affected by farm/date of slaughter (around 50% of total variance). The effect of farm/date on ash, protein, cholesterol content and beef lightness was between 20 and 30% of total variance, whereas the effect on shear force and pH was lower than 10%. 

The variability explained by individual animals was substantial for the beef content of lipids (>50% of total variance), much lower (<25%) for cholesterol and ash, and intermediate for all other beef quality traits. Lastly, the variability in ash and cholesterol content, shear force, and, in part, beef pH explained by sample/side within animal and the residual error was very high (>50%). This means that the reproducibility of these traits is lower than 50%. The highest levels of reproducibility (>90%) were for the lipid and dry matter contents of beef and for the *h** color index, while the other beef quality traits had intermediate levels of reproducibility (65 to 85%). 

### 3.2. Correlations between the Detailed Mineral Profile of Beef and Animal Performance 

Table 3 summarizes the correlations between the animals’ age at slaughter, carcass weight, and carcass gain on one side, and the detailed mineral profiles of the beef samples and their latent factors on the other. Two types of correlation were calculated: Those among the effects of farm/date (groups of animals reared on the same farm and slaughtered on the same date) and those among the individual animal within farm/date groups. With few exceptions, the farm/date and animal correlations had the same sign and similar magnitudes, although the farm/date correlations were less often significant because of their fewer degrees of freedom. 

Given the high numbers of the minerals and the productive and qualitative traits that were correlated, when farm/date and animal correlations have the same sign and similar magnitudes, even though only one may be statistically significant, in reporting the results of this survey we will simply say that the two traits are correlated, without further specification.

Age at slaughter presented modest correlations with the beef mineral profile, although the positive correlations with Na, Mg, P, Al, Ti, and Ba, and the negative correlations with K, Fe, Zn, and Sn reached the threshold of statistical significance (Table 3).

Carcass weight and carcass daily gain exhibited fewer positive correlations with the mineral contents of beef, but those with Zn and Fe (and partly with S) were particularly strong. This complex situation is more clearly summarized by the latent factors: The factor *Quantity*, the most important, was correlated only with age at slaughter (positively, i.e., unfavorably); the factor *Na + Fe + Cu* was positively correlated with carcass weight and carcass gain; the factor *K-B-Pb* was not correlated at all with any of the performance traits; the factor *Fe + Mn* was correlated negatively with age at slaughter and positively with carcass weight; lastly, factor *Zn* was highly and positively correlated with both carcass weight and carcass gain.

### 3.3. Correlations between the Detailed Mineral Profile and the Chemical Composition of Beef 

The farm/date and animal within farm/date correlations between the various minerals and the chemical composition of the beef that we analyzed are even more complex than those regarding animal performance traits. They are presented in Table 4.

Among the essential macro-minerals, only Na and P did not correlate with any of the beef chemical traits. Comparison of the beef samples from different animals within farm showed that Mg content was only modestly and negatively correlated with lipid content. Sulfur was positively correlated with the protein (and DM) content of beef, significantly when different animals were compared, non-significantly when different farm/dates were compared. Potassium was highly correlated with all chemical components (DM, protein, ash, cholesterol) except lipids. Calcium content had a strong positive farm/date correlation with the cholesterol content of beef, and modest positive animal correlations with protein, ash, and cholesterol content.

Among the essential micro-minerals, the only significant (and strong) farm/date correlations were those between Fe and Zn and the beef protein content (similar to the previously analyzed correlations with carcass weight and carcass gain). At the animal level, in addition to protein these metals were also correlated with DM, and only Zn with lipids. Mn was also correlated with DM, protein and lipids, whereas Cr was correlated (modestly) only with DM, and Cu was not correlated with any beef composition traits.

It is worth noting that several environmental micro-minerals were also related to beef composition traits. As expected for environmental minerals, some of them presented more significant farm/date correlations with beef composition (Table 4) than the other minerals. With a few exceptions, the animal correlations were of the same sign and similar magnitudes as the farm/date correlations, although they did not always reach the same level of statistical significance. Lithium was correlated negatively with lipid content and positively with ash and cholesterol contents. Boron and Pb were correlated negatively with all beef constituents, except lipids in the case of Pb, and, in the case of B, cholesterol, which, in contrast, was positively correlated with Al and Ba. Strontium was positively correlated with both cholesterol and ash. Nickel differed somewhat in being correlated negatively with the DM and lipid content of beef at the farm/date level, but positively with DM at the animal level. The fact that environmental minerals have not so far been shown to play a biological role in farm animals does not means that they are not absorbed and stored in some tissues and that these biological processes are not regulated by genetic, nutritional and health factors. The implications of the associations between some of these minerals and beef composition found here are largely unknown as there is almost nothing in the literature in this regard.

The latent explanatory factors of the beef mineral contents help simplify the analysis. The first two latent factors, which make the biggest contribution to explaining the overall mineral co-variance, showed no significant correlations with beef chemical composition. In contrast, the factor *K-B-Pb* was correlated with all chemical traits (dry matter, protein, ash, and cholesterol) except beef lipids (Table 6), the factor *Fe + Mn* with only beef protein content, and the factor *Zn* with dry matter, protein and lipids, and negatively with the ash content of beef.

### 3.4. Correlations between the Detailed Mineral Profile and the pH and Physical Properties of Beef

The farm/date of slaughter and the animal within farm/date correlations between the detailed mineral profile and the color traits of raw beef samples are summarized in Table 5, those with pH, cooking losses and shear-force of the cooked beef samples in Table 6.

Among the beef color traits, we found that lightness (*L**) was not very associated by the mineral content, with the exception of the expected negative correlation with Fe, and the positive correlations with Ti and Pb.

Moving on to the color indices, all the macro-minerals in the beef, except K and Ca, were positively correlated with *a** (redness index) and *b** (yellowness index), and also, as expected from the calculations, with *C** (chroma) and (negatively) with *h* (hue). All the essential micro-minerals, except Cr, were also positively correlated with *a** (but not *b**) and with *C**, and negatively with *h**. The associations between the environmental micro-minerals and beef color traits were more erratic: Al and Ti were favorably associated with *a*, b*,* and *C**; Li and Sn were negatively associated with the same traits; Sr was positively associated only at the farm/date level and only with *h**; while the B, Ni, and Ba variations were independent of beef color traits.

Again, the latent explanatory factors helped in the simplification process: The factor *Quantity* was correlated with all the color traits (negatively in the case of *h**); the factor *Na + Fe + Cu* was only modestly correlated with *a** and *h**; the factor *K-B-Pb* was negatively correlated with lightness, but was not correlated with beef color indices; the factor *Fe + Mn* was, as expected, correlated negatively with beef lightness, positively with and negatively with *h**; lastly, the factor Zn was correlated positively with *a** and C*, and negatively with *h**.

The relationships between the mineral content and the other beef quality traits are simpler. The acidity of beef (pH) was associated at the farm/date level only with Sr content. Tenderness (shear force) was not associated with any of the minerals in the beef. Cooking losses presented some significant correlations: Negatively (favorably) with some essential minerals (S, K and Zn), and positively with two environmental minerals (B and Pb). In this case, the latent explanatory factors do not reveal many relationships, the only significant one being the correlation between factor *K-B-Pb* and beef cooking losses (Table 6).

## 4. Discussion

### 4.1. Animal Performance and Beef Quality Traits

Animal performance and beef quality *per se* were not a primary objective of this study, but we needed a preliminary analysis of them to understand and interpret the complex relationships with the mineral profile of beef. Results show that the differences due to the animal’s breed or sex relative to age at slaughter, carcass weight, and carcass gain did not reach significance. It is worth mentioning that the beef breeds of Central Italy have a common ancestry [24], and their rearing in accordance with European Union PGI specifications for “Vitellone bianco dell’Appennino Centrale” is a traditional operation aimed at high quality production. The majority of the young bulls and heifers are reared on small-medium farms, and both the cow-calf and fattening phases have one of the highest gross margins per cow in the EU [25]. The performance traits measured in this study are very similar to those reported in a previous large survey of more than 20,000 animals [14], which also found that Chianina cattle have a greater carcass weight than the Romagnola (430 kg vs. 367 kg, respectively), as well as a greater carcass gain (0.64 kg/day vs. 0.54 kg/day) at similar ages at slaughter. 

The results for the chemical and physical traits are in the range of those reported by [26,27,28], except for the *b*, C** and *h** color parameters, which are little higher. This could be due to the fact the animals sampled for those studies came from single farms.

We have clearly shown that the effect of animal group, i.e., animals from the same farm slaughtered on the same day (farm/date effect), was the most important for almost all animal performance and beef quality traits (Figure 1), accounting for between one to two thirds of their total variance. The only exceptions were for pH, ash and cholesterol contents, and beef shear force (5 to 25% of total variance represented by farm/date). It was not the objective of this survey to analyze in detail the effects of different management and feeding practices on the PGI farms, which would have required us to sample a much larger number of farms, but rather to obtain an overview of the average productive and qualitative traits, and of their variability and relationships with the detailed mineral profile of beef. In a very large survey carried out on another Italian beef breed (Piemontese) reared in north-west Italy in accordance with another set of PGI regulations [29], the authors were able to disentangle the effect of farm from that of date of slaughter, and found that the latter was often more important than the former. They also noted that the variation between individual farms within a common beef farming system is often more important than the variation between different farming systems [30]. Our results clearly show that a first level of analysis of the relationships between beef quality and the mineral profile should be the farm/date level.

Our results also show that the variation among individual animals within farm/date group is only slightly less important than the variation between different groups of animals (Figure 1), confirming the results of [29]. This reveals the need for a second level of analysis focusing on individual animals/carcasses. The third source of variation (among different samples within animal/carcass) was generally modest, with the exception of a few traits characterized by low reproducibility (pH, ash, and cholesterol contents, and beef shear force).

### 4.2. Mineral Profile of Beef and Its Relationship with Animal Performance and Beef Quality

Similarly to what we found for beef quality traits, the mineral profile was hardly affected by the animal’s breed and sex, as the only significant differences were among breeds for the content of Ca and B, and between young bulls and heifers for the content of K, Zn, Sn, and Pb [23]. Very few of the many studies carried out on the mineral content of beef have compared different cattle breeds [31,32,33,34] or sexes [35,36,37], and most have confirmed the modest effects of these sources of variation. From these results, it seems unnecessary to study the relationships between the mineral profile and beef quality within specific breeds or sexes.

As we observed for animal performance and beef quality traits, in the case of the mineral content of beef we also found considerable variability in the relative importance of farm/date, animal/carcass within farm/date, and beef sample within animal/carcass. In particular, farm/date was the most important source of variation for Na, Mg, P, S, Li, Al, Sn, and Ti; individual animal/carcass was the most important for Mn, B, and Sr; these two sources were equally important for Fe, Cu, Zn, and Pb; and, lastly, beef sample within animal/carcass was the most important for K, Ca, Cr, Ni, and Ba [23].

The sources of variation in the beef mineral profile, like those in animal performance and beef quality, also confirm the need for at least two levels of analysis with respect to the co-variation between these two groups of traits: The farm/date level and the animal/carcass within farm/date level. The results summarized in Table 3, Table 4, Table 5 and Table 6 show the 320 correlations (52 of them significant) between the 20 minerals and the 16 animal performance and beef quality traits at the level of farm/date effects, and another 320 correlations (101 significant) at the level of individual animal/carcass within farm/date. 

The scientific literature contains some studies that also report obtaining correlation coefficients [38], often as Pearson correlations, among the raw data without making any distinction between different levels of analysis. While our data are too numerous to facilitate understanding of the interrelationships between mineral contents and beef quality, the few data that can be found in the literature are too heterogeneous and erratic to allow the same objective to be reached.

### 4.3. Latent Factors of the Beef Mineral Profile and Their Relationships with Animal Performance and Beef Quality

The fact that the content of one mineral in beef is not independent of the content of another, and that there are certain common genetic and physiological mechanisms of absorption, storage, mobilization and excretion, complicate the analysis. In the previous study [23], we examined the 190 correlations among these 20 minerals at the farm/date level, and the 190 correlations at the animal/carcass within farm/date level, and obtained numerous high correlation coefficients within each group, both positive and negative. The farm/date correlations reflect the effect of different environmental conditions, facilities, management systems, and feeding strategies, whereas the animal correlations reflect the variability among animals in the same external conditions due to their genetics, physiology, and health.

We therefore carried out a multivariate analysis of the mineral dataset, which yielded 5 unmeasured latent explanatory factors, fully independent of each other, that summarized 69% of the co-variation in the minerals [23]. Multivariate analyses have been used in previous studies, especially for authenticating beef origin or production systems [39,40], but we have found no other studies using a factor analysis of the mineral content of beef to investigate correlations with beef quality.

The most important latent factor, “*Quantity*”, which related to the concentrations of almost half the minerals analyzed (9 out of 20), all with positive loadings except Sn, and explained 45% of their total co-variance, did not correlate well with the traits studied. Among the animal performance traits, it was positively correlated only with age at slaughter at the level of individual animals within farm/date groups. It is worth pointing out that age at slaughter also has a genetic aspect as it reflects the earliness or lateness of maturation of the animals [14], and is negatively correlated with carcass weight, carcass gain and fat deposition, which could reflect greater mineral deposition in the beef of animals that require a longer fattening period to reach the level of maturation required by the market. 

As we have seen, the only beef characteristic affected by the latent factor “*Quantity*” was beef color. The beef samples with the highest scores for this latent factor were often also those with a greater intensity of color (*a*, b**, and *C**), and greater lightness (Table 5). This cannot be attributed to the effect of myoglobin because Fe is not among the minerals characterizing this latent factor (its loading is −0.03).

Fe is one of the minerals characterizing the second latent factor, “*Na + Fe + Cu*”, which explains almost 18% of the total factor variation, and also the fourth latent factor, “*Fe + Mn*”, which explains almost 11% of factor variation. As expected, both were positively correlated with the redness index (a*), and negatively with the Hue index (H*) (Table 5), while the factor “*Fe + Mn*” was also negatively correlated with beef lightness (L*). All these effects are, of course, explained by the role Fe plays in the oxidative metabolism of the muscle [41], and particularly the constituent role it plays in myoglobin, hemoglobin, and cytochromes, which could also explain the positive association of “*Na + Fe + Cu*” with both carcass weight and carcass gain (Table 3). A combined deficiency of Fe and Cu in the diet is known to reduce the growth rate of young cattle [1], although this type of deficiency is not common. Excessive Fe intake, however, could result in the depletion of Cu in cattle and hence increase the dietary requirement of this mineral [42,43]. All these relationships were significant when individual animals within groups were compared, but not when different farm/date groups were compared. At this second level, we only found a high positive correlation (+0.64) between “*Fe + Mn*” and the crude protein content of beef (Table 4). Fe and Mn are known to interact: High levels of Fe have been shown to reduce the activity of the transporters involved in the metabolism of Fe and Mn, and to increase intestinal permeability in calves [44].

The third latent factor, “*K-B-Pb*”, explaining almost 16% of factor variation, was the most correlated with beef chemical composition (Table 4). It is worth noting that K was negatively correlated with B and Pb at both the farm/date and animal/carcass within farm/date levels [23], which explains why it has a positive loading (+0.76) in this factor, whereas B and Pb have negative loadings (−0.53 and −0.54, respectively). Moreover, K is the individual mineral most positively correlated with the chemical composition of beef, and B and Pb are the individual minerals most negatively correlated with the composition of beef (lipids excluded). The role of B and Pb in beef is not known, but the highest correlations of “*K-B-Pb*” are with beef ash content, which could be explained by the fact that K is the most abundant mineral in beef (Table 2). Moreover, K in beef is involved in muscle contraction and nerve impulses, and some enzymatic reactions [1]. The animal’s body has a very limited capacity to store K, so deficiency can develop rapidly, causing, among other things, muscle weakness [45].

The latent factor “*K-B-Pb*” also correlated negatively with L* (Table 5), and, in particular, with cooking losses (Table 6). The fact that K is the major cation in intracellular fluids, hence its importance in acid-base balance, osmotic pressure and water balance [1], could be the reason why it correlates with the loss of liquids during cooking.

Lastly, the fifth latent factor, the only one associated with just one mineral, “*Zn*”, explaining just over 10% of total factor variance, also correlated positively with dry matter, protein and lipid content, and negatively with the ash content of beef (Table 4), but it also seemed to be related to beef color (correlating positively with *a** and C*, negatively with H*, Table 5). The highest correlations showed by this latent factor were not with beef quality but with animal performance traits, particularly carcass weight and carcass gain, at the farm/date and animal/carcass levels (Table 3). These many and important correlations are testimony to the importance of Zn in several aspects of the animal’s metabolism: It is an essential component of many important metalloenzymes and also triggers other enzymatic activities that affect the metabolism of carbohydrates, proteins, lipids and nucleic acids [46]. Zn deficiency, in addition to its well-known effects on the tegumental apparatus (swollen feet, parakeratotic lesions of the skin), also impairs growth, feed intake and feed efficiency [1].

## 5. Conclusions

This study had clearly shown that the minerals in beef correlated with animal performance and beef quality traits in very complex ways. A multivariate analysis of the mineral dataset condensed into five unmeasured latent explanatory factors characterized by co-variation in one to nine minerals. Two latent factors related to Fe content (“*Na + Fe + Cu*” and “*Fe + Mn*”) were found to play a special role in beef color traits, a further two (“*K-B-Pb*” and “*Zn*”) played a special role in beef chemical composition, and only “*Zn*” in carcass weight and daily gain, and beef color. Beef cooking losses were affected by “*K-B-Pb*”, whereas beef shear force was not related to any latent factor nor any individual mineral in the beef. These latent factors simplify the picture of the relationships between the minerals and animal performance and beef quality traits, facilitating the comprehension and interpretation of the role minerals.

## Figures and Tables

**Figure 1 animals-09-01073-f001:**
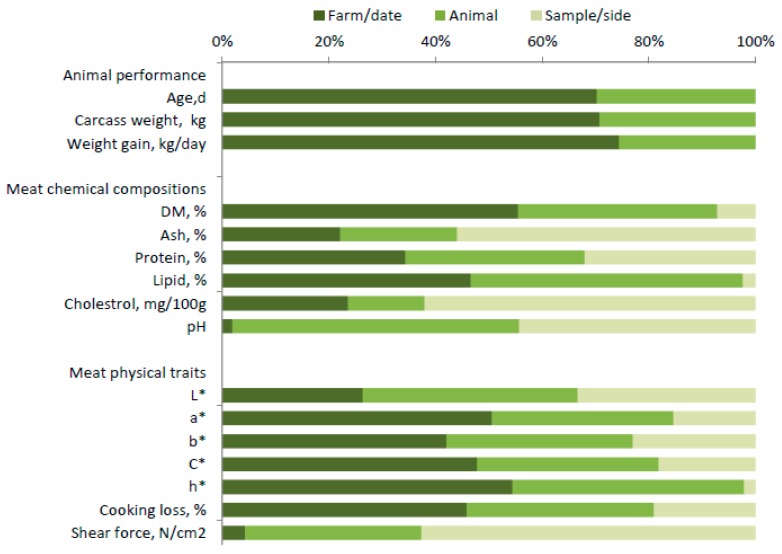
Percentage incidence of farm/date of slaughter, animal within farm/date, and sample/side within animal (residual) variances on total phenotypic variance of animal performance and beef quality traits.

**Table 1 animals-09-01073-t001:** Descriptive statistics of animal and carcass characteristics/ performance and beef composition, quality and color traits and significance of fixed breed and sex factors analyzed using a mixed model.

Traits	N ^1^	Mean	SD	Min	Max	Breed	Sex	RMSE ^2^
Animal and carcass characteristics								
Age at slaughter, d	91	701	34	589	731	-	-	21.17
Carcass weight, kg	91	403	90.7	196	572	-	-	45.74
Carcass gain, kg/d	91	0.58	0.14	0.27	0.9	-	-	0.07
Beef composition								
DM, %	177	25.9	1.7	22.7	32.6	-	*p* < 0.01	0.46
Ash, %	178	1.1	0.1	1	1.2	-	-	0.03
Protein, %	182	22.2	0.7	20.5	24.2	-	-	0.39
Lipid, %	173	2.3	1.2	0.4	8.3	-	*p* < 0.01	0.21
Cholesterol, mg/100 g	180	54.2	4.3	43.4	64.3	-	-	3.41
Beef quality traits								
pH	177	5.7	0.1	5.5	6	-	-	0.08
Cooking loss, %	179	34.6	2.8	25.9	40	*p* < 0.01	-	1.13
Shear force, (N/cm^2^)	179	30.1	6.5	14.3	49.6	*p* < 0.05	-	5.03
Beef color traits								
*L**	182	34.3	3.4	23.8	46.2	*p* < 0.05	-	1.92
*a**	182	14.3	3.2	6.3	21.2	-	-	1.38
*b**	181	13	2.4	4.6	19.3	-	-	1.28
*C**	181	19.3	3.8	8.1	28.3	-	-	1.79
*h**	180	42.2	3.9	33.5	54.4	-	-	1.42

^1^ The number of sides sampled is 182 (91 carcasses × 2 sides each) and number of data of beef quality traits lower than 182 reflect editing of data. ^2^ Root mean square error.

**Table 2 animals-09-01073-t002:** Descriptive statistics (mean and standard deviation) of the essential and environmental macro- and micro-minerals analyzed in beef samples.

Element	Minerals	Atomic Number	Category	Mean ^1^	SD ^1^
Macro-minerals, mg/kg					
Na	Sodium	11	Alkali metal	436	46
Mg	Magnesium	12	Alkaline earth metal	179	13
P	Phosphorus	15	Polyatomic nonmetal	1488	144
S	Sulfur	16	Polyatomic nonmetal	1354	78
K	Potassium	19	Alkali metal	2821	79
Ca	Calcium	20	Alkaline earth metal	46	13
Essential micro minerals, µg/kg					
Cr	Chromium	24	Transition metal	12.5	6.4
Mn	Manganese	25	Transition metal	47.6	8.1
Fe	Iron	26	Transition metal	14,260	2373
Cu	Copper	29	Transition metal	460	60
Zn	Zinc	30	Post-transition metal	39,782	5395
Environmental micro-minerals, µg/kg					
Li	Lithium	3	Alkali metal	4.1	1.7
B	Boron	5	Metalloid	164	105
Al	Aluminium	13	Post-transition metal	755	263
Ti	Titanium	22	Transition metal	14	5.1
Ni	Nickel	28	Transition metal	20.6	11.6
Sr	Strontium	38	Alkaline earth metal	47.3	20.3
Sn	Tin	50	Post-transition metal	350	154
Ba	Barium	56	Alkaline earth metal	11.2	5.9

^1^ Standard deviation on fresh basis.

**Table 3 animals-09-01073-t003:** Herd (r_H_) and animal (r_A_) correlation coefficients of animal performance with latent explanatory factors and with individual essential and environmental macro- and micro-minerals concentration in beef.

Elements	Age at Slaughter, d		Carcass Weight, kg		Carcass Gain, kg/d	
	r_H_	r_A_	r_H_	r_A_	r_H_	r_A_
Latent factors						
F1 *Quantity*	0.25	**0.31 ****	0.14	−0.04	0.05	0.05
F2 *Na + Fe + Cu*	−0.14	−0.13	0.35	**0.25 ***	0.35	**0.24 ***
F3 *K-B-Pb*	−0.17	−0.16	−0.20	−0.02	−0.12	−0.08
F4 *Fe + Mn*	−0.37	**−0.25 ***	0.40	**0.23 ***	0.44	0.19
F4 *Zn*	−0.32	−0.21	**0.78 ****	**0.62 *****	**0.74 ****	**0.64 *****
Macro-minerals						
Na	0.25	**0.24 ***	0.08	−0.01	0.01	0.06
Mg	0.18	**0.28 ****	0.09	−0.11	0.03	−0.04
P	0.22	**0.29 ****	0.15	−0.03	0.06	0.06
S	0.02	0.05	0.50	**0.32 ****	0.42	**0.39 ****
K	−0.34	**−0.21 ***	−0.07	0.10	0.03	0.05
Ca	0.35	0.19	−0.13	−0.09	−0.21	−0.04
Essential micro-minerals						
Cr	0.17	0.13	0.04	0.02	−0.01	0.06
Mn	−0.22	−0.08	0.31	0.16	0.33	0.16
Fe	−0.40	**−0.29 ***	**0.82 *****	**0.60 *****	**0.81 *****	**0.59 *****
Cu	0.37	0.19	0.21	0.01	0.09	0.05
Zn	−0.29	**−0.21 ***	**0.79 *****	**0.65 *****	**0.75 ****	**0.68 *****
Environmental micro-minerals						
Li	−0.33	−0.02	−0.29	**−0.26 ***	−0.15	**−0.31 ****
B	0.04	0.09	−0.01	−0.05	0.01	−0.04
Al	0.28	**0.22 ***	−0.04	−0.12	−0.12	−0.06
Ti	0.21	**0.21 ***	0.14	0.02	0.06	0.09
Ni	0.19	0.17	−0.37	0.04	−0.34	−0.01
Sr	0.28	0.17	**−0.61 ***	**−0.34 ****	**−0.62 ***	**−0.32 ****
Sn	−0.27	**−0.27 ****	−0.06	0.08	0.03	0.01
Ba	0.52	**0.21 ****	−0.29	−0.14	−0.38	−0.09
Pb	0.01	0.04	0.06	0.01	0.05	0.02

F1, F2, F3, F4, F5: Factor 1, 2, 3, 4 and 5; **p* < 0.05; ** *p* < 0.001; *** *p*< 0.0001.

**Table 4 animals-09-01073-t004:** Farm/date (r_F_) and animal within farm/date (r_A_) correlation coefficients of chemical composition with latent explanatory factors and with individual essential and environmental macro- and micro-minerals concentration in beef.

Elements	DM		Protein		Lipid		Ash		Cholesterol	
	r_F_	r_A_	r_F_	r_A_	r_F_	r_A_	r_F_	r_A_	r_F_	r_A_
Latent factors										
F1 *Quantity*	−0.05	−0.09	−0.15	−0.05	−0.02	−0.11	−0.38	−0.16	0.17	0.03
F2 *Na + Fe + Cu*	−0.23	0.08	−0.01	0.12	−0.26	0.01	−0.11	0.15	0.14	0.05
F3 *K-B-Pb*	0.35	**0.34 ****	**0.55 ***	**0.62 *****	0.29	0.16	**0.79 ****	**0.71 *****	**0.69 ****	**0.34 ****
F4 *Fe + Mn*	0.32	0.12	**0.64 ***	0.19	0.12	0.04	0.17	0.08	0.1	0.08
F4 *Zn*	0.48	**0.40 ****	**0.61 ***	**0.29 ***	0.37	**0.37 ****	−0.2	**−0.24 ***	−0.17	−0.07
Macro-minerals										
Na	−0.09	−0.07	−0.10	−0.13	−0.07	−0.04	−0.25	−0.09	0.3	0.12
Mg	−0.11	−0.13	−0.11	0.07	−0.09	**−0.21 ***	−0.22	0.07	0.32	0.15
P	−0.07	−0.05	−0.13	0.03	−0.05	−0.11	−0.32	−0.01	0.23	0.11
S	0.24	**0.29 ****	0.3	**0.49 *****	0.18	0.09	−0.21	0.14	0.31	0.17
K	0.24	**0.37 ****	0.51	**0.59 *****	0.2	0.17	**0.69 ****	**0.67 *****	**0.74 ****	**0.35 ****
Ca	0.03	0.12	0.26	**0.23 ***	−0.07	0.03	0.34	**0.24 ***	**0.76 ****	**0.24 ***
Ess. micro-minerals										
Cr	0.32	**0.22 ***	0.07	0.2	0.41	0.19	0.18	0.05	0.04	0.05
Mn	0.38	**0.27 ***	0.39	**0.22 ***	0.3	**0.21 ***	0.001	−0.03	0.3	−0.02
Fe	0.21	**0.26 ***	**0.53 ***	**0.31 ****	0.06	0.14	−0.26	−0.02	0.01	−0.02
Cu	−0.14	−0.16	−0.21	−0.07	−0.10	−0.16	−0.40	−0.07	0.16	0.01
Zn	0.4	**0.42 *****	**0.62 ***	**0.37 ****	0.27	**0.33 ****	−0.17	−0.13	0.03	0.01
Env. micro-minerals										
Li	−0.26	−0.19	0.16	0.05	−0.34	**−0.24 ***	**0.64 ***	**0.40 *****	0.27	**0.21 ***
B	**−0.68 ****	**−0.34 ****	**−0.61 ***	**−0.34 ****	**−0.63 ***	**−0.27 ***	**−0.54 ***	**−0.21 ***	−0.23	−0.15
Al	0.16	0.15	0.02	0.19	0.17	0.08	−0.08	0.1	0.3	**0.25 ***
Ti	0.01	−0.06	−0.22	−0.11	0.07	−0.04	−0.42	−0.17	−0.01	0.09
Ni	**−0.58 ***	**0.26 ***	−0.40	−0.06	**−0.54 ***	0.03	0.16	0.18	0.25	0.16
Sr	−0.18	−0.05	−0.13	0.05	−0.17	−0.10	0.49	**0.26 ***	**0.54 ***	**0.29 ***
Sn	0.16	0.15	0.22	0.14	0.13	0.14	0.34	0.11	−0.18	−0.10
Ba	−0.49	0.01	−0.39	0.06	−0.47	−0.06	−0.05	0.17	0.45	**0.25 ***
Pb	−0.48	**−0.26 ***	**−0.57 ***	**−0.42 *****	−0.39	−0.12	**−0.59 ***	**−0.47 *****	**−0.55 ***	**−0.38 ****

* *p* < 0.05; ** *p* < 0.001; *** *p* < 0.0001.

**Table 5 animals-09-01073-t005:** Farm/date (r_F_) and animal within farm/date (r_A_) correlation coefficients of color traits with latent explanatory factors and with individual essential and environmental macro- and micro- minerals concentration in beef.

Traits	*L**		*a**		*b**		*C**		*h**	
	r_F_	r_A_	r_F_	r_A_	r_F_	r_A_	r_F_	r_A_	r_F_	r_A_
Latent factors										
F1 *Quantity*	0.47	**0.26 ***	**0.73 ****	**0.49 *****	**0.61 ***	**0.38 ****	**0.71 ****	**0.47 *****	−0.31	**−0.23 ***
F2 *Na + Fe + Cu*	−0.09	−0.06	0.23	**0.22 ***	0.05	0.10	0.16	0.18	−0.40	**−0.29 ***
F3 *K-B-Pb*	−0.24	**−0.26 ***	−0.24	−0.08	−0.15	−0.04	−0.21	−0.07	0.19	0.04
F4 *Fe + Mn*	−0.48	**−0.22 ***	0.20	**0.22 ***	−0.06	0.03	0.10	0.15	−0.45	**−0.37 ****
F4 *Zn*	−0.53	−0.19	0.36	**0.39 ****	−0.02	0.11	0.22	**0.29 ***	**−0.68 ****	**−0.56 *****
Macro-minerals										
Na	0.34	0.12	**0.64 ***	**0.37 ****	0.48	**0.27 ***	**0.60 ***	**0.35 ****	−0.32	**−0.22 ***
Mg	0.41	0.16	**0.64 ***	**0.34 ****	0.53	**0.29 ***	**0.63 ***	**0.33 ****	−0.28	−0.15
P	0.44	0.20	**0.70 ***	**0.47 *****	**0.57 ***	**0.39 ****	**0.68 ***	**0.47 *****	−0.31	**−0.22 ***
S	0.20	0.09	**0.79 ****	**0.54 *****	**0.54 ***	**0.38 ****	**0.72 ****	**0.50 *****	**−0.54 ***	**−0.40 ****
K	−0.10	−0.13	−0.06	0.05	−0.11	0.04	−0.08	0.06	−0.05	−0.10
Ca	−0.07	−0.01	0.11	0.08	0.15	0.001	0.14	0.05	0.04	−0.16
Ess. micro-minerals										
Cr	0.31	0.13	−0.08	−0.05	0.02	0.05	−0.04	−0.01	0.11	0.14
Mn	−0.05	−0.12	**0.54 ***	**0.39 ****	0.38	0.20	0.50	**0.33 ****	−0.30	**−0.38 ****
Fe	−0.35	**−0.31 ****	**0.63 ***	**0.63 *****	0.11	0.16	0.44	**0.46 *****	**−0.92 *****	**−0.88 *****
Cu	0.23	−0.01	**0.57 ***	**0.35 ****	0.51	0.16	**0.57 ***	**0.29 ***	−0.20	**−0.35 ****
Zn	−0.44	−0.17	0.52	**0.53 *****	0.09	**0.21 ***	0.37	**0.41 *****	**−0.79 *****	**−0.65 *****
Env. micro-minerals										
Li	−0.19	−0.06	−0.38	**−0.23 ***	−0.43	−0.18	−0.42	**−0.21 ***	−0.08	0.07
B	0.11	0.16	0.16	0.01	0.09	−0.06	0.14	−0.03	−0.10	−0.05
Al	0.43	0.10	**0.63 ***	**0.36 ****	**0.65 ***	**0.35 ****	**0.67 ***	**0.38 ****	−0.08	−0.10
Ti	**0.58 ***	**0.28 ***	**0.73 ****	**0.37 ****	**0.68 ***	**0.39 ****	**0.74 ****	**0.40 *****	−0.21	−0.07
Ni	−0.06	0.15	−0.32	0.13	−0.22	−0.18	−0.29	−0.05	0.18	0.04
Sr	0.14	−0.04	−0.48	−0.17	−0.13	−0.11	−0.35	−0.15	**0.62 ***	0.11
Sn	−0.39	−0.18	**−0.62 ***	**−0.37 ****	**−0.54 ***	**−0.33 ****	**−0.62 ***	**−0.37 ****	0.24	0.14
Ba	0.34	−0.09	0.16	0.10	0.27	0.001	0.21	0.07	0.12	−0.18
Pb	0.09	**0.26 ***	−0.08	−0.08	−0.10	−0.12	−0.10	−0.11	0.03	0.03

* *p* < 0.05; ** *p* < 0.001; *** *p* < 0.0001.

**Table 6 animals-09-01073-t006:** Farm/date (r_F_) and animal within farm/date (r_A_) correlation coefficients of some quality traits with latent explanatory factors and with individual essential and environmental macro- and micro-minerals concentration in beef.

Elements	pH		Cooking Loss, %		Shear Force, N/cm^2^	
	r_F_	r_A_	r_F_	r_A_	r_F_	r_A_
Latent factors						
F1 *Quantity*	0.05	0.04	0.08	0.03	−0.32	−0.10
F2 *Na + Fe + Cu*	−0.08	0.11	−0.10	0.03	−0.09	0.01
F3 *K*-*B*-*Pb*	−0.14	0.13	−0.53	**−0.47 *****	−0.06	0.01
F4 *Fe + Mn*	0.15	0.20	−0.33	−0.11	−0.32	−0.11
F4 *Zn*	0.10	−0.07	−0.44	−0.20	−0.42	−0.20
Macro-minerals						
Na	0.01	−0.04	−0.07	0.08	−0.27	−0.02
Mg	0.01	0.06	0.01	−0.04	−0.27	−0.03
P	0.02	0.08	0.03	−0.02	−0.32	−0.06
S	−0.01	0.04	−0.16	**−0.22 ***	−0.48	−0.15
K	−0.20	0.06	**−0.61 ***	**−0.42 *****	−0.23	−0.02
Ca	−0.47	−0.03	−0.22	−0.12	−0.29	−0.10
Essential micro-minerals						
Cr	−0.15	−0.01	−0.21	−0.18	−0.11	−0.05
Mn	0.22	0.12	−0.13	−0.10	0.08	0.03
Fe	0.15	0.15	−0.36	−0.14	−0.42	−0.07
Cu	0.08	0.13	0.14	0.09	−0.06	0.12
Zn	0.08	0.02	−0.49	**−0.24 ***	−0.49	−0.19
Environmental micro-minerals						
Li	0.01	0.01	−0.46	−0.11	−0.17	−0.01
B	0.18	−0.20	0.46	**0.31 ****	0.25	−0.08
Al	0.17	0.06	−0.02	−0.18	−0.32	−0.08
Ti	0.24	0.07	0.18	0.02	−0.30	−0.09
Ni	−0.15	0.12	0.06	−0.01	0.49	−0.06
Sr	**−0.62 ***	−0.13	0.05	−0.05	−0.02	−0.01
Sn	−0.05	0.01	−0.11	−0.13	0.20	0.08
Ba	−0.36	0.00	0.16	−0.07	−0.20	−0.08
Pb	0.10	−0.20	0.53	**0.38 ****	0.30	−0.03

* *p* < 0.05; ** *p* < 0.001; *** *p* < 0.0001.

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
