# Peer review of "Relationships of a Detailed Mineral Profile of Meat with Animal Performance and Beef Quality"

_animals, 2019, doi:10.3390/ani9121073_

Round 1
Reviewer 1 Report
See attached file

Author Response
animals-645818_review
Relationships of a detailed mineral profile of meat 2 with animal performance and beef quality
Summary
Line 15. quality.
AU: Corrected
Line15-16. “Therefore, analysis of minerals in cattle´s diet is important for assessing potential effects on beef quality.”
Au: Corrected
Line 17. Use beef instead of meat, from here on.
AU: Mostly replaced with few exceptions.
Line 18. Use Hence instead of Therefore. In addition, delete “have”.
AU: Corrected
Line 20. Delete “have”.
AU: Deleted
Line 22. Delete “that indeed there”
AU: Deleted but also we have deleted “are’ to fix the sentence.
Abstract
Line 27. “Mineral profile of beef is a subject of human health interest,….”.
AU: Corrected
Line 29. After reading Material and Methods. I consider that there were only 91 samples. Do not correspond to consider each carcass side as a sample because it came from the same animal (same system, breed, management, etc). In this case each side represent a pseudo replicate.
AU: Sorry, but we are continuing to consider to have two meat samples per animals: they were collected in the same position of different sides and were processed separately (vacuum-packed, aged, transported, aliquoted, analyzed for meat quality traits, grinding freeze-dried, conserved, and analyzed for mineral contents). They would have been considered replicates if the final analysis would have been carried out twice on the same sample collected from one side. We have included in the mixed statistical model 3 random factors; the farm/date, the animal and the residual within animal (the differences between the two samples taken from each sides of every carcass). In figure 1 there is a graphical representation of the relative importance of these three random sources of variation, so readers can understand that, if we would had taken only one sample per carcass, we would have obtained a residual term equal to the sum of animal and residual within animal terms. We have not yet seen in literature papers allowing for discriminating between animal and residual within animal terms and so we feel this is a plus for this work. To avoid possible confusion we have better specified the sampling procedure and statistical model.
Introduction
Line 49. Use relevant instead of essential. Be careful, because not all mineral are essentials.
AU: Thank you for suggestion. We corrected it.
Line 53. Delete “and” (both).
AU: Done.
Line 61. Indicate example of traits that are relevant.
AU: Done
Material and Methods
Line 91. Ninety-one animals (young bulls and heifers) of the ….
AU: Corrected
Line 97. I suggest providing information of animal performance and meat composition by sex and breed. Adjusts wide of column Sex (P values) It would be interesting to perform a correspondence analysis to associate for example meat quality and origin (farm/breed). In addition, authors could also consider a cluster analysis.
AU: The experimental design was not set up to compare specifically the effect of Chianina and Romagnola breeds and of the sex on meat quality but to study the relationship between meat mineral content and beef quality traits in a real practical condition. This is why breed and sex have been included in the model together with farm/date and animal just to have a look on the relative importance of the different sources of variation. As the reviewer has seen, the number of animals per cell is not sufficient to yield affordable estimates of LSMs whereas they could be enough for quantifying the variance.
Results
Line 225. Given that there was a high variation between carcass sides, authors should consider including that in the statistical model.
AU: In a preliminary analysis we included the side (left vs right) in the model as fixed factor. As expected the effect of side was never significant on any traits analyzed (there is no reason why the left sides should be systematically different from right ones for these traits). The residual variance is based on the difference between the two samples taken for each carcass in the two sides could be mainly attributed to casual variations due to sampling procedure and heterogeneity of tissue characteristics that is typical of meat. When projecting the research we have first thought to take two samples per carcass on the same side on two consecutive ribs, but this could have led to a systematic effect of the rib position, so we finally decided to sample both sides in the same anatomical position.
Line 247. Please provide units in Table 2. In addition, Tables dimensions unfit the manuscript (too wide).
AU: Units are incorporated and table is adjusted
Line 332. Do not use the words “much affected”, instead use “very associated”
AU: Corrected
Line 343. “Again, the latent explanatory factors helped in the simplification process:”
AU: Corrected
Line 349. Delete “the before other….”
AU: Done
Discussion
Line 359. What was the power of the test for breed and sex, particularly sex? That because the “n” was low, only 8 heifers. The differences in diets (i.e. types of pastures and species) as a relevant element to consider and as a potential explanation of the correlations observed, especially given the farm / date effect, are not considered in the discussion. What potential explanation they pose for the variations between the samples of the carcasses.
AU: We agree that the power of the test for comparing the two breeds and especially the two sexes was not very high. But, as said before, this was not the main objective of the study but a mere quantification of the factors of variation in our animals sampled to represent practical conditions. This is why we have not included LSMs in the tables.

Reviewer 2 Report
Comments on the paper animals-645818 by Patel and co-workers sent for publication to Animals.
The paper intends to study the relationships between minerals in meat with animal performance and beef quality. The paper is interesting and clarifies some unknowns in this topic. The paper is also well written. However, a preprint of this paper (doi:10.20944/preprints201911.0025.v1) exists and the authors didn’t cite it.
My comments are hereafter that would improve the quality of this work.
The referee don’t understand why there is a summary (line 14) and then an abstract (line 27)? Please clarify otherwise combine both.
Line 28: please give the abbreviation of ICP-OES.
Line 89: can the authors check that it is the SEUROP scale that was used instead of EUROP?
Line 91: how the statistical analyses were performed with the few 8 heifers? It is unbalanced compared to the others?
Line 95: add a reference at the end of the sentence.
Line 97 for Table 1: it is unclear why in a descriptive data the authors are giving RMSE? Also, the breed and sex had no sense in this table.
The referee is wondering if the term “animal performance” is appropriate. I suggest animal and carcass characteristics.
Also, for Table 1 several variables are only presented with abbreviations. Please highlight those variables with footnotes at the bottom of the Table.
Please for hue angle, write it as “h*” and not in capital letter. For all color traits, they need to be in italic.
Line 118: how the temperature was controlled at the core of the steak. The time of 40 min is very high, and the large literature suggests 30 min. please, cite the reference that you used.
Line 155: describe how? And are the data homogenous or no?
Line 171: explain further how the residuals were managed and the reference that you followed.
Line 174-179: the referee suggests standardising the data and then computing the correlation coefficients, otherwise they are not accurate. Please, carefully check this major comment. This will allow you retain only the robust correlations that deserve attention.
Line 181: the referee doesn’t understand how the same data are analysed for another paper that is not published. This needs to be avoided. Please include those results here or share the paper to check that there is not self-plagiarism and redundancy to this paper.
Line 247: why the range of the values is not given in this table?
I suggest to the authors to highlight the different groups (breed and sex) for the variables used with Principal Component Analyses.
Author Response
REVIEWER 2:
Comments on the paper animals-645818 by Patel and co-workers sent for publication to Animals.The paper intends to study the relationships between minerals in meat with animal performance and beef quality. The paper is interesting and clarifies some unknowns in this topic. The paper is also well written. However, a preprint of this paper (doi:10.20944/preprints201911.0025.v1) exists and the authors didn’t cite it.
AU: Please consider that preprint is an option offered and encouraged by this journal to anticipate some information to possible readers and it should not be intended as a duplicate publication as it will be removed after acceptance and publication of the manuscript.
My comments are hereafter that would improve the quality of this work.
AU: We thank the reviewer for his interest and time dedicated to the improvement of our paper. We really appreciated it.
The referee don’t understand why there is a summary (line 14) and then an abstract (line 27)?
Please clarify otherwise combine both.
AU: This is because Animals ask for both. It is a part of journal format and an essential item in order to submit the article.
Line 28: Please give the abbreviation of ICP-OES.
AU: Provided
Line 89: can the authors check that it is the SEUROP scale that was used instead of EUROP?
AU: the initial EUROP griddle for muscularity evaluation was modified in the SEUROP griddle by the EEUU many years ago for including carcasses from double muscled breeds. Since then only SEUROP griddle is accepted, independently of the fact that some S carcass is available or not (like in the present trial).
Line 91: how the statistical analyses were performed with the few 8 heifers? It is unbalanced compared to the others?
AU: We agree that the sex ratio is unbalanced but, as said in response to the similar comment by reviewer 1, the sex ratio represent the proportions often found in the beef market as the majority of heifers are retained by farmers to replace the culled cows. The effects of breed and sex are not the main objective of this study, that was to analyze relationships between meat mineral content and beef quality, and their inclusion in the model has the objective of representing and quantifying the sources of variation affecting meat traits.
Line 95: add a reference at the end of the sentence.
AU: Instead of a reference we included the web-site address of the Herd Book.
Line 97 for Table 1: it is unclear why in a descriptive data the authors are giving RMSE? Also, the breed and sex had no sense in this table.
AU: In an earlier version of this manuscript we included two tables: one with the descriptive statistics and the other with the ANOVA. Later we decided to suppress the second table including in the first the significance of the fixed factoris and RMSE (as a measure of residual variability) and to extrapolate the data on proportion of variances of the random factors in a graphical form (Figure 1) hoping to increase readability of the manuscript.
The referee is wondering if the term “animal performance” is appropriate. I suggest animal and carcass characteristics.
AU: Accepted
Also, for Table 1 several variables are only presented with abbreviations. Please highlight those variables with footnotes at the bottom of the Table.
AU: Changed
Please for hue angle, write it as “h*” and not in capital letter. For all color traits, they need to be in italic.
AU: Changed
Line 118: How the temperature was controlled at the core of the steak. The time of 40 min is very high, and the large literature suggests 30 min. please, cite the reference that you used.
AU: We have used a procedure often used in Europe (now included in the reference list). The time of 40 min warrant that the 70°C temperature is achieved in the center of all samples.
Line 155: describe how? And are the data homogenous or no?
AU: We checked normality and homogeneity of data using histogram, Q-Q probability plot and by Shapiro-Wilks test. The text has been changed.
Line 171: explain further how the residuals were managed and the reference that you followed.
AU: we have modified the text.
Line 174-179: the referee suggests standardising the data and then computing the correlation coefficients, otherwise they are not accurate. Please, carefully check this major comment. This will allow you retain only the robust correlations that deserve attention.
AU: Please consider that, differently from regression coefficients (not considered here) correlation coefficients are not modified by data standardization and that accuracy of standardized data depends on the accuracy of raw data. In any case, the inclusion in the tables of the significance level of each correlation coefficient should be sufficient to inform the readers about the confidence to give to the estimates.
Line 181: the referee doesn’t understand how the same data are analyzed for another paper that is not published. This needs to be avoided. Please include those results here or share the paper to check that there is not self-plagiarism and redundancy to this paper.
AU: we guarantee for no self-plagiarism risk. The other paper, now accepted for publication, do not regards meat quality traits. It is a “methodological” paper on the analysis of minerals in meat. In that paper there is the detailed description of the analytical procedure adopted for the quantification of minerals, the statistical analysis of the variability sources of minerals (not included here) and the multivariate analyses of relationships among the minerals (not with meat quality like here) leading to the identification of the mineral latent factors. Here the latent factors were analyzed in relation to meat quality. To avoid overlapping between the two papers, here the analyses of minerals, not being the objective of the paper, was summarized and for details the readers are addressed, correctly, to the other paper. To further avoid possible confusion the paragraph 3.2 (Mineral contents of beef and their latent explanatory independent factors) have been now deleted and the pertinent information transferred to Material and method (2.3.2. Mineral analysis). Also table 2 have been moved to material and methods. So the only data in common with the previous paper are the means and SD of minerals, here presented as material of this paper. The ANOVA model was the same, but here it was used to analyzed meat traits and not mineral contents. No information or data is in common in the results section and also the discussion is obviously focused here on meat quality and relations of minerals with beef quality and not on minerals per se. Also conclusions have been modified.
Line 247: why the range of the values is not given in this table?
AU: see the answer to the previous comment.
I suggest to the authors to highlight the different groups (breed and sex) for the variables used with Principal Component Analyses.
AU: Please see the previous comments.
